# Early Detection of SARS-CoV-2 Omicron BA.4 and BA.5 in German Wastewater

**DOI:** 10.3390/v14091876

**Published:** 2022-08-25

**Authors:** Alexander Wilhelm, Shelesh Agrawal, Jens Schoth, Christina Meinert-Berning, Daniel Bastian, Laura Orschler, Sandra Ciesek, Burkhard Teichgräber, Thomas Wintgens, Susanne Lackner, Frank-Andreas Weber, Marek Widera

**Affiliations:** 1Institute for Medical Virology, University Hospital, Goethe University Frankfurt, Paul-Ehrlich-Str. 40, D-60596 Frankfurt, Germany; 2Institute IWAR, Water and Environmental Biotechnology, Technical University of Darmstadt, Franziska-Braun-Str. 7, D-64287 Darmstadt, Germany; 3Emschergenossenschaft/Lippeverband, Kronprinzenstraße 24, D-45128 Essen, Germany; 4Ruhrverband, Kronprinzenstraße 37, D-45128 Essen, Germany; 5FiW e.V., Research Institute for Water Management and Climate Future at RWTH Aachen University, Kackertstraße 15-17, D-52056 Aachen, Germany; 6German Center for Infection Research (DZIF), D-38124 Braunschweig, Germany; 7Fraunhofer Institute for Translational Medicine and Pharmacology ITMP, Theodor Stern Kai 7, D-60595 Frankfurt am Main, Germany; 8Institute of Environmental Engineering, RWTH Aachen University, Mies-van-der-Rohe-Strasse 1, D-52074 Aachen, Germany

**Keywords:** COVID-19 surveillance, SARS-CoV-2 monitoring, wastewater-based epidemiology (WBE), Omicron, BA.4, BA.5, variant of concern

## Abstract

Wastewater-based SARS-CoV-2 epidemiology (WBE) has been established as an important tool to support individual testing strategies. The Omicron sub-variants BA.4/BA.5 have spread globally, displacing the preceding variants. Due to the severe transmissibility and immune escape potential of BA.4/BA.5, early monitoring was required to assess and implement countermeasures in time. In this study, we monitored the prevalence of SARS-CoV-2 BA.4/BA.5 at six municipal wastewater treatment plants (WWTPs) in the Federal State of North Rhine-Westphalia (NRW, Germany) in May and June 2022. Initially, L452R-specific primers/probes originally designed for SARS-CoV-2 Delta detection were validated using inactivated authentic viruses and evaluated for their suitability for detecting BA.4/BA.5. Subsequently, the assay was used for RT-qPCR analysis of RNA purified from wastewater obtained twice a week at six WWTPs. The occurrence of L452R carrying RNA was detected in early May 2022, and the presence of BA.4/BA.5 was confirmed by variant-specific single nucleotide polymorphism PCR (SNP-PCR) targeting E484A/F486V and NGS sequencing. Finally, the mutant fractions were quantitatively monitored by digital PCR, confirming BA.4/BA.5 as the majority variant by 5 June 2022. In conclusion, the successive workflow using RT-qPCR, variant-specific SNP-PCR, and RT-dPCR demonstrates the strength of WBE as a versatile tool to rapidly monitor variants spreading independently of individual test capacities.

## 1. Introduction

The severe acute respiratory syndrome coronavirus type 2 (SARS-CoV-2) is the causative agent of the coronavirus disease 2019 (COVID-19). SARS-CoV-2 was identified in December 2019 in the Chinese metropolis of Wuhan and rapidly evolved into a global pandemic [1,2]. During the evolutionary process, nucleotide polymorphisms might arise in the SARS-CoV-2 genome, potentially affecting the viral transmissibility and the susceptibility to monoclonal antibodies (mAb) and antibodies from convalescent as well as vaccine-elicited sera [3]. In recent months, several lineages of the SARS-CoV-2 Omicron variant (B.1.1.529), including BA.1 and BA.2, have prevailed over the preceding Delta variant [4]. Compared to the predecessor variants, BA.1 and BA.2 were shown to exert increased immune escape to naturally acquired or vaccine-elicited neutralising antibodies [5,6,7,8,9]. The two most recent variants, BA.4 and BA.5, have high similarity to BA.2 and carry specific mutations in the spike protein, including Δ69/70, L452R, and F486V, while Q493 is found in the preceding variants BA.1 and BA.2 [10,11]. In particular, substitutions in spike position 452 (L452R, L452Q, L452M) are associated with a high reproduction number. BA.4 and BA.5 were shown to severely escape the antibodies elicited by the early Omicron infection, indicating advanced resistance to previous immunity [12,13]. Furthermore, the early published data report a more efficient spread in lung cells and the potentially higher pathogenicity of BA.4/BA.5 when compared with BA.2 [11]. Hence, due to the severe immune escape, high transmissibility, and unknown pathogenic potential, early monitoring of BA.4/BA.5 is required to consider the regulation of local countermeasures.

Using molecular reverse transcriptase quantitative polymerase chain reaction (RT-qPCR) analysis the detection of SARS-CoV-2 gene fragments in wastewater was shown to correlate with the COVID-19 prevalence in the catchment area of the sewage treatment plant [14,15]. A wide variety of advanced detection methods, including RT-qPCR, digital (droplet) PCR (d(d)PCR), and genomic sequencing (NGS), were described to detect and quantify variant-specific RNA fragments [16,17,18,19,20,21,22]. For all the approaches, key substitutions of the novel variant, such as K417N for Omicron BA.1, were shown as suitable targets for molecular detection [16].

In this study, we demonstrate a step-wise coordinated workflow using the advantages of multiple PCR detection methods for the rapid tracking of the SARS-CoV-2 Omicron BA.4/BA.5 variant. Using an already available L452R-specific PCR assay and a variant-specific single nucleotide polymorphism PCR targeting E484A/F486V, we describe a rapid and cost-effective stepwise monitoring PCR approach, which might be applicable to the monitoring of future pandemics by adjusting the primer and probes.

## 2. Materials and Methods

### 2.1. Sewage Sampling

Wastewater samples (*n* = 72) were collected at six municipal wastewater treatment plants (WWTPs): Klaeranlage Emschermuendung (KLEM), Dortmund-Scharnhorst (DoS), Dortmund-Deusen (DoD), Bottrop (BOT), Dinslaken (DIN), and Duisburg-Alte-Emscher (DAE), all located in North Rhine-Westphalia (NRW)/Germany (Figure 1). The key properties of the WWTPs operated by the public German water board Emschergenossenschaft and Lippeverband are summarized in Table 1**.** Flow-proportional 24 h composite samples were collected after the grit chamber at the WWTP inlet using an installed autosampler. With the aim of a practical application, the settings of the installed automatic samplers of each WWTP were adopted from routine testing. In most cases, 12 successive samples were collected in 2-h intervals. The pumping time was set depending on the amount of wastewater for each WWTP.

### 2.2. Sample Processing and RT-qPCR Quantification of Viral RNA

Sample processing and RT-qPCR quantification of the viral RNA was performed as described previously [16]. Accordingly, 100 mL of each sample was filtered through electronegative membrane filters (0.45 µm pore size, Merck Millipore, Darmstadt, Germany) at a pressure of 6 bar using Nitrogen gas. After filtration, the filter was cut and placed into Lyses Tubes J (Analytik Jena, Jena, Germany). After addition of 1 mL DNA/RNA shield reagent (Zymo Research Europe, Freiburg im Breisgau, Germany), the samples were processed in a speed mill (SpeedMill plus, Analytik Jena) at room temperature and 50 Hz for 2 min and, subsequently, centrifuged at 10,000× *g* for another 2 min. The total RNA of the samples was isolated using the automated purification system InnuPure C16 touch (Analytik Jena, Jena) and the innuPREP AniPath DNA/RNA Kit IPC16 according to the manufacturer’s instructions. Accordingly, 400 µL of lysate was used for extraction and the total RNA was eluted in a volume of 100 µL using RNAse-free water. Subsequently, 5 µL of RNA was used for one-step RT-qPCR analysis performed in duplicates. The remaining RNA was stored at −80 °C.

The initial screening for the presence of the Omicron variant was performed at the cooperative laboratory of Emschergenossenschaft and Lippeverband in Essen, which is operated together with Ruhrverband, by analysis of the L452R mutation using the QuantiNova Pathogen +IC Kit (Qiagen, Hilden, Germany) on a qTOWER^3^ real-time-thermocycler (Analytic Jena, Jena, Germany) according to the manufacturers’ instructions. Data analysis was performed using qPCRsoft version 4.1.3.0 (Analytic Jena, Jena, Germany). In addition to the detection of the SARS-CoV-2 spike (N1/N2, amplicon length 72 bp and 67 bp, respectively), the Pepper Mild Mottle Virus (PMMoV) was used as a human fecal control (primer and probes provided by the manufacturer IDEXX). The L452R assay (amplicon length 106 bp) specificity and sensitivity for the detection of L452R-carrying SARS-CoV-2 variants were evaluated using inactivated authentic SARS-CoV-2 stocks derived by viral outgrowth assays and patient-derived swab samples (Appendix A).

All the non-proprietary primer and probes used in this study are provided in Table 2. The performance of the detection assay was comparable to that of other labs, as evaluated in multiple interlab comparison studies.

### 2.3. Sample Processing for Digital PCR-Based SARS-CoV-2 Variant Detection and Quantification of Viral RNA

The wastewater samples were processed using an optimized workflow based on the previously described 4S method [24]. Forty milliliters of the wastewater samples was poured into a 50 mL tube containing 9.35 g sodium chloride and 400 µL of TE buffer (1 M Tris, 100 mM EDTA, pH 7.2). The samples were agitated until all the sodium chloride was dissolved and heat-inactivated at 70 °C for 45 min. Each sample was filtered twice through a 5 µM filter and mixed with 70% ethanol in a 1:1 ratio. Subsequently, the RNA was isolated using an adapted protocol of the Wizard Enviro TNA Kit (Promega, Walldorf, Germany). The RNA was eluted with 50 µL RNA-free water for subsequent analysis. The QIAcuity OneStep Advanced Probe Kit (Qiagen, Hilden, Germany) was used and the reaction was performed on a QIAcuity Digital PCR System (Qiagen, Hilden, Germany) using QIAcuity Nanoplate 26k 24-well plates (Qiagen, Hilden, Germany). Ten microliters of RNA was used for each reaction, performed in a 40 µL volume per reaction with two technical replicates. QIAcuity Software Suite_version 1.2.18 (Qiagen, Hilden, Germany) was used for the data analysis. The primer and probe sequences are described in Table 2. The remaining RNA was stored at −80 °C. The raw data of the dPCR analysis and further details on the data analysis and partitioning are available on request.

### 2.4. SARS-CoV-2 Variant-Specific Single Nucleotide Polymorphism PCR (SNP-PCR)

After initial screening for the presence of L452R, a variant-specific single nucleotide polymorphism PCR targeting E484A/F486V was performed to confirm BA.4/BA.5. SARS-CoV-2 Omicron variants BA.2, BA.4, BA.5 harbour the E484A mutation, but F486V is a unique mutation found in BA.4/BA.5. The proprietary primer and probes (SARS-CoV-2 VirSNiP Mutations Assay, Cat.-No. 53-0839-96) were purchased by TIB Molbiol Syntheselabor GmbH (Berlin, Germany). After cycling, a melting curve analysis was performed using a heating rate of 0.2 °C/second. A high melting peak (~60 °C) is an indication of the presence of the SARS-CoV-2 variants BA.4/BA.5, while BA.2 sequences yield a lower melting temperature peak (~53 °C) (Appendix A).

### 2.5. Specificity Testing Using Authentic SARS-CoV-2 Isolates

Nasopharyngeal swab-derived authentic SARS-CoV-2 isolates and original sample material were used for specificity testing, as described previously [9,16]. Briefly, swab material was used for the viral outgrowth assays, using Caco2 and A549-AT cells [25,26]. All cell culture work involving infectious SARS-CoV-2 was performed under biosafety level 3 (BSL-3) conditions. Sample inactivation for further processing was performed with previously evaluated methods [27]. RNA from the temperature-inactivated SARS-CoV-2 was diluted and subsequently used for assay validation. To this end, we used a multiplex approach based on the simultaneous use of two differently labelled locked nucleic acid (LNA)-containing probes (Table 2) and performed allelic discrimination analysis (Appendix A). The sequences of SARS-CoV-2 isolates used in this study are available at the NCBI Genbank repository: MT358638, MT358641 (parental B); MT358643 (parental B.1); MZ427280 (Alpha, B.1.1.7), MW822592 (Beta, B.1.351); MZ315141 (Delta, B.1.617.2), OL800702 (Omicron, BA.1), OM617939.1 (Omicron, BA.2), OP062266 (Omicron, BA.4), OP062267, OP062268 (Omicron, BA.5).

### 2.6. Epidemiological Data

The epidemiological data on SARS-CoV-2 cases in NRW, including the variant-specific portions in Germany, were obtained from the official data repository of the federal Robert Koch Institute (RKI), which is in charge of public health surveillance.

### 2.7. Next Generation Sequencing Analysis for Confirmation of the SARS-CoV-2 Omicron BA.4 and BA.5 Subvariants

For the next-generation sequencing (NGS) analysis, 200 mL of the untreated wastewater samples was concentrated using 100-kDa Centricon Plus-70 centrifugal ultrafilters (Merck). The RNA was extracted from the concentrate using the MagMAX Microbiome Ultra Nucleic Acid isolation kit (Thermo Fisher Scientific, Waltham, MA, USA) according to the manufacturer’s protocol. SARS-CoV-2 sequencing was performed on the Ion Torrent platform according to a previous study [17]. Briefly, the method allows the obtaining of nearly full coverage of the SARS-CoV-2 genome while generating amplicons ranging between lengths of 125–275 bp. Sequencing was performed using an Ion Torrent 540 chip on an Ion S5 sequencer (Thermo Fisher Scientific). The data analysis was performed directly on the Ion Torrent sequencer using the installed Ion Torrent Suite v5.12.2. Using the TMAP software included in the Ion Torrent Suite, the sequences were aligned to a SARS-CoV-2 reference genome (Wuhan-Hu-1 [GenBank accession numbers NC_045512 and MN908947.3]) and single-nucleotide variants (SNVs) were called using Variant Caller v5.16.0.5 with default parameters.

## 3. Results

### 3.1. Wastewater Surveillance Allows Early Detection of SARS-CoV-2 Omicron Variants BA.4/BA.5

Wastewater samples were collected at six municipal WWTPs located in North Rhine-Westphalia (NRW)/Germany (Figure 1). To monitor the occurrence of BA.4/BA.5, we used a primer/probe set that was already available for the detection of SARS-CoV-2 Delta (Appendix A). The assay was validated using several inactivated SARS-CoV-2 variants of concern (VoCs) and patient-derived swab samples for subsequent RT-qPCR analysis in a decentral laboratory near the WWTPs.

Wastewater samples were initially monitored for PMMoV used as a human fecal marker and the total SARS-CoV-2 RNA load (N1/N2) by RT-qPCR, showing an increasing viral load in all observed WWTP beginning on 1st June 2022 (Figure 2A). Concomitantly to this increase, lower ct values for the BA.4/BA.5 characteristic substitution L452R were observed using a target-specific PCR assay yielding the FAM signal for L452R and the Cy5 signal for L452, starting with a peak on the 23rd of May in the wastewater treatment plant Dinslaken (DIN) and on the 25th of May for the wastewater treatment plants in Dortmund-Deusen (DoD), Dortmund-Scharnhorst (DoS), Emschermuendung (KLEM), and Duisburg-Alte-Emscher (DAE) and on the 1st of June in Bottrop (BOT) (Figure 2B). A preceding increase of the ct values for L452 could already be observed on 18 May 2022 at most of the wastewater treatment plants. At DoS, this increase was observed with a delay of 5 days on 23 May 2022 (Figure 2B). A first rough estimation of the relative proportions of the L452R mutant based on the reciprocal ct values for L452/L452R revealed a more than 50% L452R fraction on 25 May 2022 for DoD and DAE, on 30 May 2022 for DoS, and on 1 June 2022 for KLEM, DIN, DAE, and BOT (Figure 2C). However, for the treatment plants DoD and DAE, a strong drop was observed in the follow-up measurement, which may be due to the non-absolute copy number determination of the RT-qPCR method.

This first evidence of the non-exclusive BA.4/BA.5 substitutions (L452R) was further subjected to a variant-specific single nucleotide polymorphism PCR targeting E484A/F486V.

### 3.2. Confirmation of Omicron BA.4/BA.5 by Variant-Specific Single Nucleotide Polymorphism PCR

To confirm that the L452R-based detection definitely originated from SARS-CoV-2 BA.4/BA.5, a variant-specific single nucleotide polymorphism PCR, which was available for the identification of BA.4 or BA.5 variants from nasopharyngeal swabs, was performed. A preceding validation of the assay also approved its use for the detection of substitution in wastewater samples (Appendix A). Subsequently, consecutive samples collected at several sites were measured with this assay. The melting curve analysis of the amplicon products clearly confirmed the presence of a variant carrying E484A/F486V, which is exclusively found in BA.4/BA.5 (Figure 3). In agreement with the RT-qPCR-derived data, the comparison of the peaks of both E484A (present in BA.1/2) and E484A/F486V (specific for BA.4/BA.5) [28] revealed a shift towards E484A/F486V confirming the presence of BA.4/BA.5. A concomitant decrease of the E484A-specific peak confirmed this assumption.

### 3.3. Tracking the Mutation Fraction Using Digital PCR (dPCR)

As the assay for L452R was already predesigned to detect the SARS-CoV-2 Delta variant and evaluated for high specificity and sensitivity for BA.4/BA.5 (Appendix A), this target was used as an applicable surrogate marker to track the course of the mutant fractions over time.

As shown in Figure 4A, only a minor difference when comparing the national and the NRW-specific 7-day incidence was observed during the study period (Figure 4A, left panel). Following individual testing, Omicron BA4./5 became the dominant variant in Germany on a national scale on 12th July 2022 (Figure 4A, right panel).

The dPCR-derived findings suggest that the Omicron variants BA.4/BA.5 occurred in the WWTP catchment area in a rapidly rising fraction (>40%) on the 25th of May in both the KLEM and the DoD WWTPs (Figure 4B). The decrease of the L452-specific signal clearly indicated the displacement of SARS-CoV-2 BA.2.

Ahead of the available public health data at this time point, the relative increase in the mutant fraction over time could be quantitatively monitored, confirming Omicron as the predominant variant on 26 June 2022. In agreement with the public health data, the dPCR-based quantification of total SARS-CoV-2 RNAs (N1/N2) precisely resembled the 7-day incidence. Of note, the strong increase in total SARS-CoV-2 RNA measured with an N1- and N2-specific assay strongly correlated with the dominance of L452R. Our data on the current BA.4/BA.5 monitoring demonstrate the strength of the successive workflow using RT-qPCR, variant-specific SNP-PCR, and RT-dPCR.

### 3.4. Confirmation of SARS-CoV-2 Omicron BA.4 and BA.5 by Next Generation Sequencing

In order to confirm the presence and to monitor the relative proportion of the BA.4 and BA.5 Omicron subvariants individually, the wastewater samples from DoD and KLEM were sequenced from three different time points. During the acute variant replacement phase, a PCR-assay for the discrimination of both SARS-CoV-2 Omicron subvariants, BA.4 and BA.5, was not available. Hence, only the NGS sequencing enabled a discrimination between the two subvariants and revealed BA.5 as the majority and BA.4 as the minority variant.

The fraction of the read abundance of the L452R mutation was also analyzed and is associated with BA.4 and BA.5 (Figure 5A). In a sample obtained from DoD on 16 May 2022, around 10% of the total SARS-CoV-2 read abundance was allocated to BA.4 and BA.5. By 7 June 2022, BA.4 and BA.5 constituted around 50% of the total fraction in the DoD and KLEM samples. Overall, from the middle of May to the end of June, an increase of 1 log10 in the read abundance of L452R mutation was observed (Figure 5B). Furthermore, the change was monitored in the read abundance of the mutations unique for BA.4 and BA.5, respectively (Figure 5C), i.e., D3N in M protein for BA.5 and P151S in N protein for BA.4. The reported dominance of BA.5 was based on the high abundance of the D3N mutation in all the samples. BA.4 was between 30–36% in the KLEM samples and 16% in the DoD sample from 28 June 2022 (Figure 5C). These data agreed with the allele frequency of the D3N and P151S mutations in those samples (Figure 5D). The allele frequency of D3N increased significantly over the sampling time in comparison to the allele frequency of the P151S mutation.

The incidence and variant prevalence of BA.2, BA.4, and BA.5 in Germany during the study period, as reported by the RKI (Appendix A), was congruent with the NGS-derived data. With more than 90%, BA.5 was the pre-dominant variant in calendar week 29, 2022, while BA.4 was detectable in less than 5% of the cases. However, NGS analysis demonstrates regional differences (36% and 16% BA.4 in KLEM and DoD, respectively) in the spread and dynamics of the individual subvariants (Figure 5).

## 4. Discussion

The data from the wastewater-based SARS-CoV-2 epidemiology (WBE) were shown to correlate with the COVID-19 prevalence in the catchment area of the respective sewage treatment plants [14,15]. Depending on the WWTP, the catchment area, and the specific ratio of commuters or permanent residents, the WBE may even provide a reporting time advance of several days compared to the clinical findings. Hence, WBE can effectively support individual testing strategies and becomes even more important in times of declining testing willingness. In addition, WBE might be used to monitor the emergence of novel variants of concern (VoC) already at the early stages. Previous studies have shown that measuring the concentration of variant-specific SARS-CoV-2 RNA in wastewater can efficiently track regional variant dynamics [16,29,30,31].

In our study, WWTPs from a densely populated metropolitan area were investigated (Figure 1, Table 1). Nevertheless, within this study, which was conducted as part of the COVIDready project located in NRW in Germany, we included several representative wastewater treatment plants of different sizes. These range from 56,000 to 906,222 connected inhabitants (Table 1). However, compared to the more rural areas, even the smallest WWTP is still relatively large. The samples were collected during routine operation and no changes were made to the routine operation. Of note, commuters might reside during their work days in metropolitan areas, generating a regionally disproportionate high load in the wastewater, while, concomitantly, a correspondingly lower input would result in rural areas.

WBE is an inexpensive and non-invasive mass surveillance method compared to individual testing and thus represents a suitable surveillance complement in low- and middle-income countries [32]. Although centralized wastewater management systems might be limited in low-income settings, in contrast to the high costs of NGS sequencing, cost efficient RT-PCR-driven variant tracking might be more applicable [31,32].

The SARS-CoV-2 variant Delta was predominant until the end of 2021 but was rapidly displaced [29]. In a study performed by the COVIDready consortium in December 2021, we were able to identify the occurrence of the novel SARS-CoV-2 variant Omicron (BA.1) in wastewater from North Rhine-Westphalia (Germany) using a fast and efficient decentralized workflow [16]. The hallmark of this workflow was that previously existing detection assays, such as those for the detection of the K417N substitution originally validated for the detection of the Beta variant, were immediately available for the detection of new variants harboring the identical substitution. Subsequently, the confirmation and tracking of the variant was performed by NGS sequencing and digital PCR, respectively. Importantly, K417N was present in the Omicron BA.1 and BA.2 subvariants, but not in the previously dominant Delta variant. Hence, by simultaneous detection of the parental variant, a continuous transition of both variants could be tracked over time and quantified by digital PCR, not only in a relative but also in an absolute manner [16]. An initial screening using already available PCR assays allows early monitoring; however, careful evaluation of the assays for specificity is highly recommended. Many commercial PCR assays are able to detect the newly emerging variant, but may give false-positive signals for other variants [16]. For individual testing, these weak false-positive signals are negligible, as the sample usually contains a single variant. The wastewater matrix, however, is much more complex, as the sum of all the variants within the catchment area of a WWTP is covered simultaneously. The WWTP KLEM, for example, is connected to more than 900,000 nominal residents (Table 1).

The current detection of BA.4/BA.5 using a PCR assay detecting L452R originally developed for the detection of previous variants but not for targeting Omicron BA.1 and BA.2, again proved the effectiveness of the decentral workflow described above [16]. Initially, the primers and probes originally designed for the detection of the SARS-CoV-2 Delta-specific L452R substitution were rapidly validated using inactivated authentic viruses and evaluated for their suitability to detect BA.4/BA.5 (Appendix A). In addition to the detection of characteristic but not exclusive mutations, the newly emerging variant needs to be confirmed and identified; this has been performed so far by cost- and time-intensive sequencing methods [33,34]. In this study, the initial workflow was further developed, and the verification of the emerging SARS-CoV-2 variant was replaced by a rapidly available, inexpensive, and excellent performing single nucleotide polymorphism PCR (SNP-PCR). Compared to the preceding variant, only a few substitutions in the spike gene, including L452R and F486V, might be used for discrimination. Using a variant-specific SNP-PCR targeting E484A/F486V, we confirmed the presence of SARS-CoV-2 Omicron BA.4/BA.5. Because of their rapid availability, variant-specific melting curve-based PCRs have proven to be a suitable method for rapid confirmation [35]. Of note, the time course was in excellent agreement with the dynamics quantified via dPCR on the basis of the L452R ratios. However, a limitation of this approach is the lack of discrimination capacity between BA.4 and BA.5. Hence, in order to finally distinguish BA.4 from BA.5, a variant-specific LNA assay targeting non-spike mutations (e.g., D3N or P151S in SARS-CoV-2 M and N, respectively) might be feasible. Alternatively, the exclusion of the previous variants by using already established assays might be applicable, although giving only an indirect proof.

Further approaches, such as the allele-specific RT-qPCR assay published by Lee and colleagues, are available [29]. This particular assay detects a stretch of spike mutations (493 to 498) that are suitable for discriminating Omicron BA.1 and BA.2 from other variants. In contrast to the LNA-based assay developed in our study, the allele-specific assay relies on a variant-specific forward primer, while the fluorescence signal is derived from a variant-independent probe [29].

Irrespective of the assay design, PCR genotyping allows real-time tracking and detection of circulating variants; however, it has limitations compared to NGS sequencing, which is able detect the whole genome even of unknown variants without the need of predesigned amplicon targets.

The incidence and variant prevalence of SARS-CoV-2 Omicron BA.2, BA.4, and BA.5 in NRW and Germany during the study, as reported by RKI, was congruent with the NGS-derived data. With 92.1%, BA.5 was the pre-dominant variant in calendar week 29, 2022, while BA.4 was detectable in less than 4.8%. Of note, these public health data were not available during the study period and might be biased due to the strongly declining testing willingness in Germany.

We conclude that a coordinated PCR-based workflow might serve as a robust and sensitive early warning system in pandemic control. We recommend implementing such a coordinated workflow in wastewater-based epidemiology as a complementary measure in addition to individual testing strategies. Using this established workflow synergistically with NGS, we will be able to rapidly adapt to and track the spread and dynamics of new SARS-CoV-2 variants and future pandemics.

## Figures and Tables

**Figure 1 viruses-14-01876-f001:**
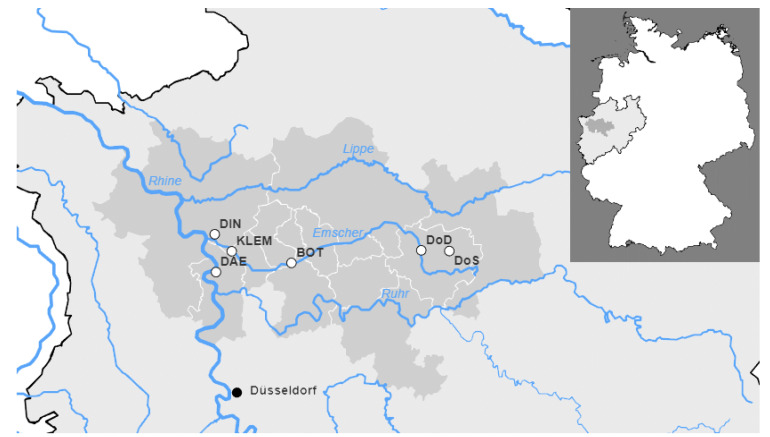
The wastewater treatment plants monitored as part of this study. Regional representation of the municipal districts (dark gray) connected to the respective wastewater treatment plants in the German state of North Rhine-Westphalia. The sewage treatment plants are shown as white circles with black borders. Rivers are shown in blue, federal state borders in black. Cities are shown as black circles for further orientation. A general map of Germany to illustrate the overall view, indicating the colors, is given at the top right.

**Figure 2 viruses-14-01876-f002:**
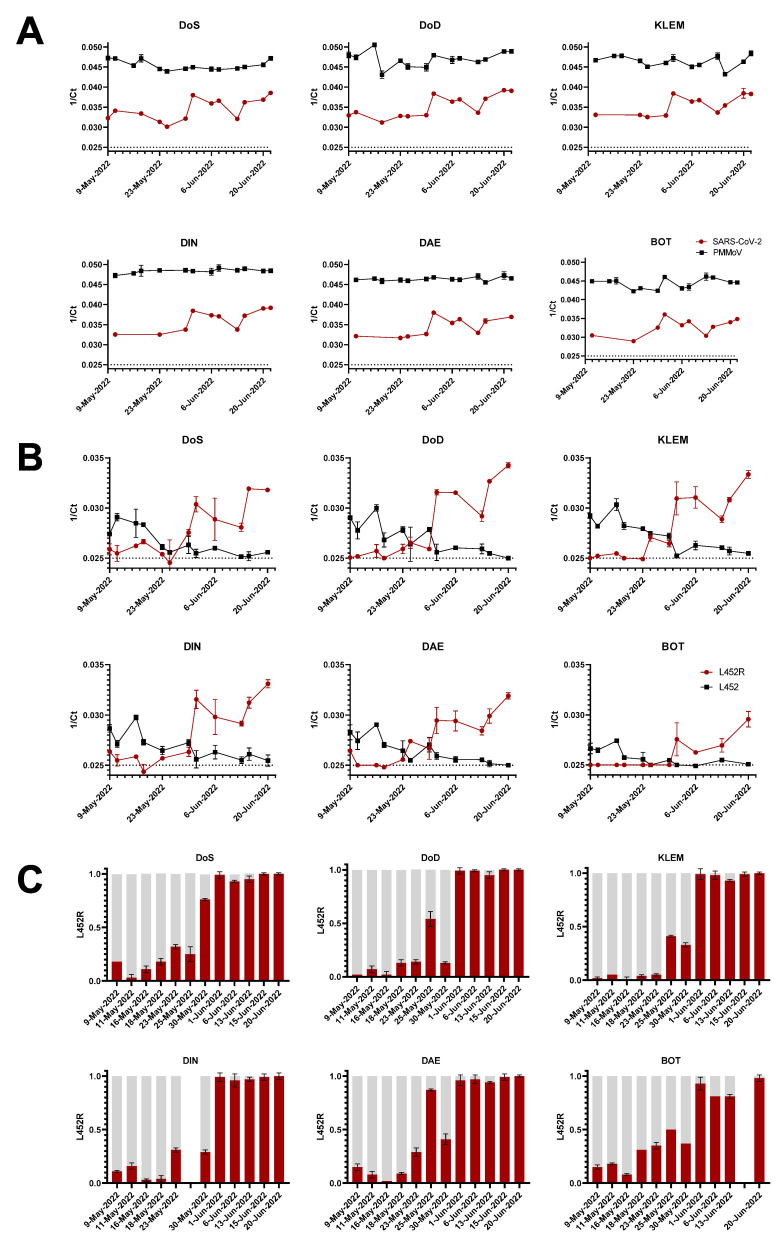
Monitoring of SARS-CoV-2 viral RNA fragments in six different wastewater treatment plants. Wastewater samples (single samples per day) were analyzed by RT-qPCR in two technical replicates for (**A**) the presence of SARS-CoV-2 viral load (red) using N1/N2 targets (simultaneously detected in a dual target assay), PMMoV (black line), and (**B**) Omicron BA.4/BA.5 characteristic mutation L452R in the SARS-CoV-2 spike. The corresponding reciprocal ct values (1/ct) are illustrated for each WWTP. The dotted line indicates the limit of detection (>ct = 40). (**C**) Red (L452R) and grey (L452) bars represent the ratio between BA.4/BA.5 and non-BA.4/BA.5 fractions estimated by calculating the ct values for both targets. For technical reasons, no data are available for DIN from 25 May 2022 and for BOT from 15 June 2022.

**Figure 3 viruses-14-01876-f003:**
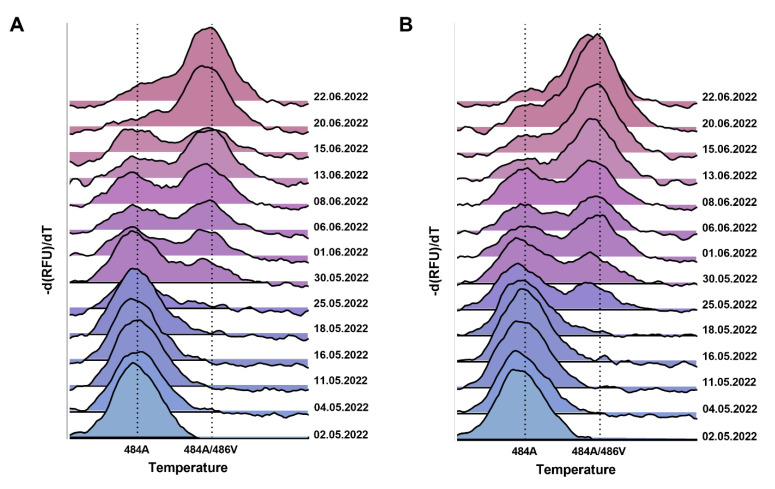
Confirmation of E484A/F486V substitution by melting curve PCR analysis. A variant-specific single nucleotide polymorphism PCR (SNP-PCR) targeting E484A/F486V was performed using RNA derived from the two WWTPs (**A**) KLEM and (**B**) DoD. The peak on the left (E484A) shows the specific melting temperature of an amplicon containing E484A, representing non-BA.4/BA5 SARS-CoV-2 RNA. Due to the low binding caused by mismatch, a lower temperature is required compared to E484A/F486V with a more efficient probe match. The color scheme is for orientation only and does not correlate with the proportion of the mutant.

**Figure 4 viruses-14-01876-f004:**
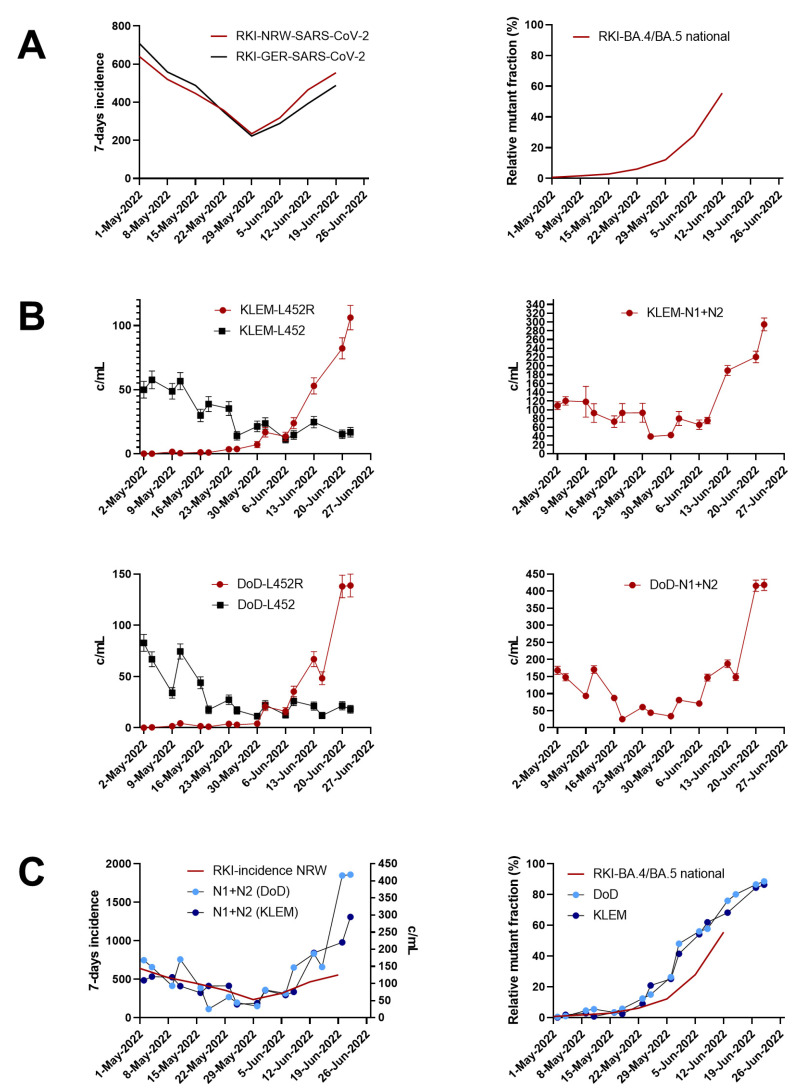
Tracking of SARS-CoV-2 BA.4/BA.5 specific mutant fraction of L452R using digital PCR. (**A**) Seven-day incidence (left panel) and the SARS-CoV-2 Omicron BA.4/BA.5 mutant fraction (right panel) for NRW and Germany are indicated as available during the study period. Epidemiological data are based on individual testing and were obtained from official data repository of the German federal Robert Koch Institute (RKI). (**B**) SARS-CoV-2 Omicron BA.4/BA.5 and non-BA.4/BA.5 detection targeting the substitution L452R (left panel) for WWTPs DoD and KLEM. Total SARS-CoV-2 levels in WWTPs DoD and KLEM were quantified using a SARS-CoV-2-specific N1/N2 assay (right panel) (**C**) Overlay of the 7-day incidence in NRW compared to detected SARS-CoV-2 levels in wastewater of WWTPs DoD and KLEM (left panel). Overlay of the relative mutant fraction of SARS-CoV-2 Omicron BA.4/BA.5 as provided by RKI compared to the relative fraction of L452R determined in wastewater using RT-dPCR (right).

**Figure 5 viruses-14-01876-f005:**
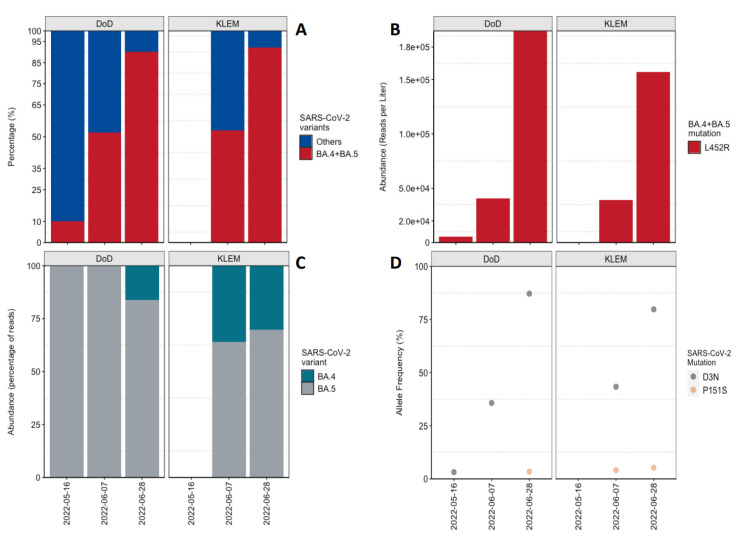
Sequencing of wastewater samples for confirmation of SARS-CoV-2 BA.4 and BA.5 subvariants. (**A**) Percentage abundance of BA.4 and BA.5 together in comparison to the other SARS-CoV-2 variants based on the read abundance of the L452R spike protein mutation. (**B**) Plot showing the increase in the read abundance of the L452R mutation, which is associated with BA.4 and BA.5 during the sampling period. (**C**) Percentage abundance of BA.4 and BA.5 based on their unique mutations, i.e., D3N for BA.5 and P151S for BA.4. (**D**) Allele frequency of the D3N and P151S mutations detected in the samples.

**Table 1 viruses-14-01876-t001:** Key properties of the six wastewater treatment plants sampled in this study. Data were obtained from ELWAS-WEB, an electronic water management system for administration in NRW (accessed 29 June 2022).

WWTP	Acronym	Nominal Number of Connected Residents	Population Equivalent	Annual Wastewater Flow in 2020 [m^3^/a]
Emschermuendung	KLEM	906,222	426,173	348,703,426
Dortmund-Scharnhorst	DoS	113,439	45,342	12,192,038
Dortmund-Deusen	DoD	399,425	185,144	47,716,171
Bottrop	BOT	732,816	678,818	131,203,662
Duisburg Alte Emscher	DAE	242,172	133,535	35,807,933
Dinslaken	DIN	56,812	19,157	3,812,870

**Table 2 viruses-14-01876-t002:** Sequences of non-proprietary primers and probes used for SARS-CoV-2 detection. PMMoV, pepper mild mottle virus; IDT, purchased from Integrated DNA Technologies. “+N” indicates LNA positions. FAM, 5′ 6-FAM (Fluorescein) modification; ZEN, internal quencher for fluorescence-quenched probes (IDT). 3IABkFQ, 3′ Iowa Black FQ quencher; 3IAbRQSp, 3′ Iowa Black RQ quencher; Cy5, 5′ Cy5 fluorescence dye.

Primer/Probe	Target Gene	Sequence (5′-3′)	Source
N1 probe	SARS-CoV-2 N	FAM/ACCCCGCAT/ZEN/TACGTTTGGTGGACC/3IABkFQ/	[23]
N2 probe	SARS-CoV-2 N	FAM/ACAATTTGC/ZEN/CCCCAGCGCTTCAG/3IABkFQ/	[23]
N1 fwd	SARS-CoV-2 N	GACCCCAAAATCAGCGAAAT	[23]
N1 rev	SARS-CoV-2 N	TCTGGTTACTGCCAGTTGAATCTG	[23]
N2 fwd	SARS-CoV-2 N	TTACAAACATTGGCCGCAAA	[23]
N2 rev	SARS-CoV-2 N	GCGCGACATTCCGAAGAA	[23]
L452R fwd	SARS-CoV-2 S	CTTGATTCTAAGGTTGGTGGTAAT	IDT, this study
L452R rev	SARS-CoV-2 S	CGGCCTGATAGATTTCAGTTG	IDT, this study
L452R probe 1	SARS-CoV-2 S	Cy5/TA+C+C+T+GTATA+G+ATTG/3IAbRQSp	IDT, this study
L452R probe 2	SARS-CoV-2 S	FAM/TAC+C+G+GTA+TA+G+AT/3IABkFQ	IDT, this study

## Data Availability

All sequences of SARS-CoV-2 variants used for assay validation are available at NCBI GenBank under the following accession numbers: MT358638, MT358641, MT358643, MZ427280, MW822592, MZ315141, OL800702, OM617939.1, OP062266, OP062267, OP062268, NC_045512, and MN908947.3.

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
