# Peer review of "Early Detection of SARS-CoV-2 Omicron BA.4 and BA.5 in German Wastewater"

_viruses, 2022, doi:10.3390/v14091876_

Round 1

Reviewer 1 Report

viruses-1848629. Early Detection of SARS-CoV-2 Omicron BA.4/5 in German 2 wastewater

In this study, Wilhelm and co-workers monitored the prevalence of SARS-CoV-2 BA.4/5 at six municipal wastewater treatment plants (WWTPs) in the Federal State of North-Rhine-Westphalia in May and June 2022. Initially, they used L452R-specific primers/probes originally designed for SARS-CoV-2 Delta detection. Later on, the presence of BA.4/5 was confirmed by variant-specific single nucleotide polymorphism PCR targeting E484A/F486V.

General comments.

Even if the topic is of interest, however, we believe that discriminating BA.4/5 from BA.2 is preliminary but incomplete data. Currently, it is clear that we are in a variant replacement phase, but we need to understand if BA.4 or BA.5 is going to replace completely BA.2. Therefore, we believe that detecting only BA.4/5 is partially useful. We suggest additional testing of BA.4/BA.5 samples aimed atdistinguishing the two subvariants. BA.4 and BA.5 are indistinguishable in the spike protein; however, they have key distinctive mutations in other regions of the genome that could be used for subvariant specific testing by real time or sequencing.

A description of the variant prevalence in Germany during the study period (BA.2 vs BA.4 vs BA.5) could be useful in the introduction or discussion section.

It is not clear how many samples were tested in this work; moreover, a list of all primers/probes used in this study should be included in a Table. In page 3 line 114 the author state that Primer and probe sequences are described in Supplementary Table 1. However, we only see an appendix at the end of the manuscript with “scheme 1”; but, we believe that primers and probe should be better described in the main manuscript, and not in appendix

There are two sub-paragraphs (2.2 and 2.3) with the same title “Sample processing and RT-qPCR quantification of viral RNA”, but a completely different description. This is not clear. Were the wastewater samples processed as described in 2.2 from 100ml or as described in 2.3 from 40 ml? The number of wastewater samples collected from the six WWTPs is to be included.

The rationale behind the E484A/F486V testing is not immediately clear, since F486V is a key mutation of BA.4/BA.5, but E484A is shared by Omicron, Ba.2, Ba.4, Ba.5, and also Ba.2.75. We suggest explaining in 2.4 (SARS-CoV-2 variant-specific single nucleotide polymorphism PCR (SNP-PCR)) what is detailed in the legend of figure 3 (discrimination based on specific melting temperature of the amplicon) to make the concept clear to the reader immediately.

Specific comments:

-          Page 2, lines 48-50. “The two most recent variants BA.4 and BA.5 have high similarity to BA.2 but carry exclusive mutations including Δ69/70; L452R, F486V, and Q493 in the spike protein”. While Δ69/70; L452R, and F486V are all specific mutations carried by BA.4 and BA.5, it is not correct to state that Q493 is a specific mutation for BA.4/5 (instead, Q493R is a specific mutation of BA.2). Please correct.

-          Page 2, line 52 “In particular, substitutions in Spike L452R/Q/M”, please explain

-          Page 2, lines 64-65. “substitutions absent in the predecessing variant”: the sentence should be better modified with “key substitutions of the novel variant” instead of “absent in the previous”

-          Page 2, line 68. “pre-existing L452R-PCR”, please explain, give reference and primers/probes

-          Page 2, line73. Please add the number of samples investigated in this work

-          Page 3 line 95. Please add the volume of extracted RNA, as well as the volume of RNA used for testing

-          Page 3, lines 125-126. Even if there is a reference to this sentence, we suggest giving some details, at least the number of samples and the variants (VOCs and/or VOIs tested)

-          Page 3, lines 128-129. Even if there is a reference to this sentence, we suggest giving some details, at least the principle of inactivation (temperature? UV? Other?)

-          Page 4, lines 136-139. This is already described in materials and methods can be deleted.

-          Page 5, line 173 “originally developed for individual testing”, please explain

-          Page 8, line 217. Please explain “decentralized wastewater”.

-          Page 10 line 290. “scheme 1”. Should be supplementary Figure 1?

Author Response

Reviewer#1:

In this study, Wilhelm and co-workers monitored the prevalence of SARS-CoV-2 BA.4/5 at six municipal wastewater treatment plants (WWTPs) in the Federal State of North-Rhine-Westphalia in May and June 2022. Initially, they used L452R-specific primers/probes originally designed for SARS-CoV-2 Delta detection. Later on, the presence of BA.4/5 was confirmed by variant-specific single nucleotide polymorphism PCR targeting E484A/F486V.

General comments.

Even if the topic is of interest, however, we believe that discriminating BA.4/5 from BA.2 is preliminary but incomplete data. Currently, it is clear that we are in a variant replacement phase, but we need to understand if BA.4 or BA.5 is going to replace completely BA.2. Therefore, we believe that detecting only BA.4/5 is partially useful. We suggest additional testing of BA.4/BA.5 samples aimed at distinguishing the two subvariants. BA.4 and BA.5 are indistinguishable in the spike protein; however, they have key distinctive mutations in other regions of the genome that could be used for subvariant specific testing by real time or sequencing.

Response: We thank the reviewer for the helpful evaluation of our manuscript. We agree that discrimination of the two SARS-CoV-2 subvariants is of interest, although the required primer probes were not available for testing during the relevant sampling campaign. We have therefore added the retrospective NGS-sequencing analysis for discrimination of BA.4 and BA.5 now shown in Figure 5. This data is agreement with the public health surveillance data, which is now also available as Supplementary Figure 2. We added a corresponding passage into the main text and updated the discussion section.

A description of the variant prevalence in Germany during the study period (BA.2 vs BA.4 vs BA.5) could be useful in the introduction or discussion section.

Response: We added a supplementary figure (Supplementary Figure 2) showing the variant prevalence in Germany during the study period (BA.2 vs BA.4 vs BA.5). Here, congruent with the NGS-related data, it becomes clear that BA.5 plays a dominant role with 92.1%, and BA.4 only a minor role with 4.8% (calendar week 29). Of note, this public health data was not available during the study period and might be biased by limited test willingness in Germany. We also added a corresponding discussion passage.

It is not clear how many samples were tested in this work; moreover, a list of all primers/probes used in this study should be included in a Table. In page 3 line 114 the author state that Primer and probe sequences are described in Supplementary Table 1. However, we only see an appendix at the end of the manuscript with “scheme 1”; but, we believe that primers and probe should be better described in the main manuscript, and not in appendix

Response: For variant detection, a single sample with two technical PCR replicates was used for subsequent analysis. We added this information to the material and methods section and figure legends. Available (non-proprietary) sequences of primers and probes used for SARS-CoV-2 detection are now listed in table 2. However, sequences for PMMoV-detection as well as for targeting E484A/F486V were purchased by IDEXX and TIB Molbiol Syntheselabor GmbH (Berlin, Germany), respectively, and are not available.

There are two sub-paragraphs (2.2 and 2.3) with the same title “Sample processing and RT-qPCR quantification of viral RNA”, but a completely different description. This is not clear. Were the wastewater samples processed as described in 2.2 from 100 ml or as described in 2.3 from 40 ml? The number of wastewater samples collected from the six WWTPs is to be included.

Response: We thank the reviewer for pointing at this issue. We apologise for the confusion we might have caused and now specified sections 2.2 and 2.3. Section 2.3 describes the sample processing for digital PCR-based SARS-CoV-2 variant detection and quantification of viral RNA performed in the central laboratory in Frankfurt am Main, Germany / Hesse.

The rationale behind the E484A/F486V testing is not immediately clear, since F486V is a key mutation of BA.4/BA.5, but E484A is shared by Omicron, Ba.2, Ba.4, Ba.5, and also Ba.2.75. We suggest explaining in 2.4 (SARS-CoV-2 variant-specific single nucleotide polymorphism PCR (SNP-PCR)) what is detailed in the legend of figure 3 (discrimination based on specific melting temperature of the amplicon) to make the concept clear to the reader immediately.

Response: We included additional information and now explain the rationale behind the E484A/F486V testing. In particular, SARS-CoV-2 Omicron variants BA.2, BA.4, BA.5 harbour the F486V mutation, but F486V is a unique mutation found in BA.4/BA.5. A high melting peak (~60°C) is an indication of the presence of the SARS-CoV-2 variants BA.4/BA.5 while BA.2 sequences yields a lower melting temperature peak (~53°C).

Specific comments:

-          Page 2, lines 48-50. “The two most recent variants BA.4 and BA.5 have high similarity to BA.2 but carry exclusive mutations including Δ69/70; L452R, F486V, and Q493 in the spike protein”. While Δ69/70; L452R, and F486V are all specific mutations carried by BA.4 and BA.5, it is not correct to state that Q493 is a specific mutation for BA.4/5 (instead, Q493R is a specific mutation of BA.2). Please correct.

Response: We corrected the sentence, accordingly.

-          Page 2, line 52 “In particular, substitutions in Spike L452R/Q/M”, please explain

Response: We rephrased the sentence, accordingly.

-          Page 2, lines 64-65. “substitutions absent in the predecessing variant”: the sentence should be better modified with “key substitutions of the novel variant” instead of “absent in the previous”

Response: Thank you for the suggestion. We corrected the sentence, accordingly.

-          Page 2, line 68. “pre-existing L452R-PCR”, please explain, give reference and primers/probes

Response: We rephrased the sentence and now refer to table 2.

-          Page 2, line73. Please add the number of samples investigated in this work

Response: We added the number of analysed samples (n=72).

-          Page 3 line 95. Please add the volume of extracted RNA, as well as the volume of RNA used for testing

Response: We added the missing information.

-          Page 3, lines 125-126. Even if there is a reference to this sentence, we suggest giving some details, at least the number of samples and the variants (VOCs and/or VOIs tested)

Response: We added the requested information.

-          Page 3, lines 128-129. Even if there is a reference to this sentence, we suggest giving some details, at least the principle of inactivation (temperature? UV? Other?)

Response: We evaluated several methods and identified heat inactivation (>90°C) as the most applicable. Details were included, accordingly.

-          Page 4, lines 136-139. This is already described in materials and methods can be deleted.

Response: We deleted the redundant details.

-          Page 5, line 173 “originally developed for individual testing”, please explain

Response: We rephrased the sentence.

-          Page 8, line 217. Please explain “decentralized wastewater”.

Response: We rephrased the sentence and now highlight the strength of the successive workflow using RT-qPCR, variant-specific SNP-PCR, and RT-dPCR.

-          Page 10 line 290. “scheme 1”. Should be supplementary Figure 1?

Response: We changed the caption to “Supplementary Figure“.

Reviewer 2 Report

Thank you for submitting your manuscript to Viruses. While the overall approach of your work and the sampling plan/locations are ideal for the presented study, I have a few issues with the absence of any viral internal control. Using abundant human fecal markers have been widely accepted by researchers and institutions like the US-CDC and the European Commission to help normalizing data in light of sometimes highly variable wastewater inflow into WWTP. Furthermore, the authors should consider stating if the MIQE guideline was followed to ensure that results can be compared with confidence in between sampling dates and operators.

Besides those points, it is recommended that the authors increase the number of references in the discussion and ensure that their local case study is discussed in an international context. In this light, I would also ask the authors to compare the presented approach to similar methods that have been frequently presented and adopted during the pandemic.

Minor comments and remarks can further be found in the attached PDF.

Author Response

Reviewer#2:

Thank you for submitting your manuscript to Viruses. While the overall approach of your work and the sampling plan/locations are ideal for the presented study, I have a few issues with the absence of any viral internal control.

Using abundant human fecal markers have been widely accepted by researchers and institutions like the US-CDC and the European Commission to help normalizing data in light of sometimes highly variable wastewater inflow into WWTP.

Response: The current findings on accompanying parameters of our wastewater treatment plants show that surrogate virus gene copies in particular are subject to high fluctuations even independent of volume flow. Accordingly, a surrogate virus normalisation of the data based on e.g. PMMoV is under debate in our consortium. This is also in line with Langeveld et al (2021), where a volume flow-based approach to normalisation is preferred. However, we agree that short-term fluctuations in surrogate virus gene copies may possibly provide a clue for estimating single-value quality. We have therefore provided the PMMoV ct-values in n figure 1, but refrain from normalising our data to them.

Furthermore, the authors should consider stating if the MIQE guideline was followed to ensure that results can be compared with confidence in between sampling dates and operators.

Response: We added additional information following the dMIQE2020 checklist into the material and method section. We also added a statement that raw data will be available on request.

Besides those points, it is recommended that the authors increase the number of references in the discussion and ensure that their local case study is discussed in an international context. In this light, I would also ask the authors to compare the presented approach to similar methods that have been frequently presented and adopted during the pandemic.

Response: As suggested, we now significantly extended the discussion section and added more references.

Minor comments and remarks can further be found in the attached PDF.

Response: Thank you for the helpful suggestions for improvement, which we have incorporated into the manuscript.

Reviewer 3 Report

The manuscripts describes the detection of Omicron BA.4/5 in sewage collected from different WWTP in NRW, Germany. The manuscript is well written. Study design is appropriate and data are clearly presented. The authors clearly demonstrate that sewage is excellently suited to detect variants of concern at early stage by 'simple' RT-PCR techniques. They also show that data obtained by WBE correlate with data obtained from individual testing. Such findings are important for the public and should help to implement WBE to standard pathogen monitoring.

I have only one minor remark. Please specify sections 2.2 and 2.3. It is not clear for me which samples were processed with which method and why there are differences in e.g. sample processing. 

Author Response

Reviewer#3:

The manuscripts describes the detection of Omicron BA.4/5 in sewage collected from different WWTP in NRW, Germany. The manuscript is well written. Study design is appropriate and data are clearly presented. The authors clearly demonstrate that sewage is excellently suited to detect variants of concern at early stage by 'simple' RT-PCR techniques. They also show that data obtained by WBE correlate with data obtained from individual testing. Such findings are important for the public and should help to implement WBE to standard pathogen monitoring.

I have only one minor remark. Please specify sections 2.2 and 2.3. It is not clear for me which samples were processed with which method and why there are differences in e.g. sample processing.

Response: We thank the reviewer for the evaluation of our manuscript. We apologise for the confusion we might have caused and now specified sections 2.2 and 2.3.

“2.3: Sample processing for digital PCR-based SARS-CoV-2 variant detection and quantification of viral RNA”

Round 2

Reviewer 1 Report

The Authors have addressed all of concerns regarding the original manuscript.

We suggest minor revisions to be included in the text:

The title and the abstract should be modified including major modifications of the revised manuscript (sequencing data discriminating BA.4 from BA.5).

Line 62. Please, use the same description BA.4/BA.5 (or BA.4/5) throughout the text

Line 76. Please check the order of Table. Table 2 should be cited after table 1.

Line 77-78. It is not clear how this approach can be applicable to monitor future pandemics.  

Lines 150-151. Duplication, already described in lines 147-148.

Author Response

Thank you for the effort to work through the manuscript again. We have corrected the open points accordingly:

The title and the abstract should be modified including major modifications of the revised manuscript (sequencing data discriminating BA.4 from BA.5).

>>> We changed the title as suggested 

Line 62. Please, use the same description BA.4/BA.5 (or BA.4/5) throughout the text

>>> We now use the same description (BA.4/BA.5)

Line 76. Please check the order of Table. Table 2 should be cited after table 1.

>>> We removed the first reference to Table 2 in the introduction, which was no essential.

Line 77-78. It is not clear how this approach can be applicable to monitor future pandemics.

>>>We have now added that the method can also be used for future pandemics or pathogen detection by adapting the primers and probes.  

Lines 150-151. Duplication, already described in lines 147-148.

>>> Thank you, we corrected accorrdingly.

Reviewer 2 Report

Thank you for addressing all my concerns and remarks, the manuscript should be a valuable contribution to the ongoing scientific discours and published in its current form.

Author Response

Thank you for your effort to work through the manuscript again.